# Unraveling How Tumor-Derived Galectins Contribute to Anti-Cancer Immunity Failure

**DOI:** 10.3390/cancers13184529

**Published:** 2021-09-09

**Authors:** Diego José Laderach, Daniel Compagno

**Affiliations:** 1Molecular and Functional Glyco-Oncology Laboratory, IQUIBICEN-CONICET, Buenos Aires C1428BGA, Argentina; danielcompagno@qb.fcen.uba.ar; 2Departamento de Química Biológica, Facultad de Ciencias Exactas y Naturales, Universidad de Buenos Aires, Buenos Aires C1428EGA, Argentina; 3Departamento de Ciencias Básicas, Universidad Nacional de Luján, Luján L6700, Argentina

**Keywords:** galectins, tumor immune evasion, cancer immunotherapy, lymphocyte homeostasis

## Abstract

**Simple Summary:**

This review compiles our current knowledge of one of the main pathways activated by tumors to escape immune attack. Indeed, it integrates the current understanding of how tumor-derived circulating galectins affect the elicitation of effective anti-tumor immunity. It focuses on several relevant topics: which are the main galectins produced by tumors, how soluble galectins circulate throughout biological liquids (taking a body-settled gradient concentration into account), the conditions required for the galectins’ functions to be accomplished at the tumor and tumor-distant sites, and how the physicochemical properties of the microenvironment in each tissue determine their functions. These are no mere semantic definitions as they define which functions can be performed in said tissues instead. Finally, we discuss the promising future of galectins as targets in cancer immunotherapy and some outstanding questions in the field.

**Abstract:**

Current data indicates that anti-tumor T cell-mediated immunity correlates with a better prognosis in cancer patients. However, it has widely been demonstrated that tumor cells negatively manage immune attack by activating several immune-suppressive mechanisms. It is, therefore, essential to fully understand how lymphocytes are activated in a tumor microenvironment and, above all, how to prevent these cells from becoming dysfunctional. Tumors produce galectins-1, -3, -7, -8, and -9 as one of the major molecular mechanisms to evade immune control of tumor development. These galectins impact different steps in the establishment of the anti-tumor immune responses. Here, we carry out a critical dissection on the mechanisms through which tumor-derived galectins can influence the production and the functionality of anti-tumor T lymphocytes. This knowledge may help us design more effective immunotherapies to treat human cancers.

## 1. Introduction

In humans, the immune system is constituted by approximately 10^13^ T lymphocytes at a given time [1]. However, lymphoid homeostasis is highly dynamic, depending on the continuous production of naïve cells in central hematopoietic organs, their activation, survival, and generation of immune memory in peripheral organs. Each of these processes depends on continuous biological signals that lymphocytes receive from their microenvironment [2]. At a molecular level, multiple signaling pathways are involved in lymphocyte regulation. Although we only have a partial understanding of this subject, an extensive compendium of the literature supports galectins as being prominent actors in the regulation of immune homeostasis in both central and peripheral organs (as discussed below).

Currently, 12 human galectin members have been described and found in GenBank (https://www.ncbi.nlm.nih.gov/genbank/; accessed on 15 July 2021). Galectins are involved in the regulation of different cellular processes, among which we can mention cell differentiation, cell adhesion and migration; gene transcription and RNA splicing; and cell cycle and apoptosis [3,4]. Such biological functions depend on their lectinic properties (recognition of N-acetyllactosamine sequences displayed on the cell surface, in the extracellular matrix, or intracellular glycoconjugates) (Table 1). Besides, and studied to a lesser extent, they also depend on non-lectinic properties mediated by interactions of galectins with nucleic acids, proteins, lipids, and complex biomolecules (reviewed in [4], Table 1).

Interestingly, the expression profile of galectins is altered during several pathological conditions (expert opinions for each disease have been reviewed in [5,6,7,8,9,10,11,12,13,14]). In particular, the production of galectins-1, -3, -7, -8, and -9 is deregulated during tumorigenesis [15,16,17]. These galectins are detected in transformed cells and their associated stroma in various types of cancers [18,19]. In the latter case, it is not clear whether stromal cells produce galectins by themselves or take them up from neighboring transformed cells [20,21,22]. However, galectins play essential roles in controlling the tumor microenvironment properties independently of their cellular origin [18,19]. Notably, tumor-associated hypoxia and inflammation contribute to the deregulation of galectins expression [23,24,25,26]. As a consequence, increased levels of these galectins are detected in the sera of cancer patients. Until mid-2021, more than 55 independent studies have been recorded in this field (here, and due to space limitations, we only cite some examples on each galectin member [27,28,29,30,31,32,33,34,35,36,37,38]).

Increased galectins production in cancers has been proposed as a prognostic factor since they can easily be detected in the biological fluids and their expression in the tumor microenvironment generally predicts a poor clinical outcome for patients [15,19,39,40,41]. Besides being used as interesting biomarkers, these tumor-derived galectins display active roles in defining tumor progression. Despite the multiple influences these proteins have on the behavior of tumor cells (e.g., tumorigenesis, metastasis, and angiogenesis) [16], it is clear that they constitute an important strategy applied by tumors to evade an immune attack [42]. This review aims to deepen our knowledge on how galectins, abundantly produced by tumors, can impact the production, activation, and effector function of anti-tumor T lymphocytes; some of these processes occur at anatomic sites distant from the tumor (Figure 1).

### 1.1. Do Circulating Tumor-Derived Galectins Have Any Impact on Naïve T Cell Production?

From the approximately 2–7 × 10^7^ naïve T cells produced in the thymus every day during adulthood [122,123], very few cells are specific against self-components due to the relative efficiency of the negative selection process [124]. However, if the cancer immunosurveillance theory is correct (which is supported by the low frequency of tumor occurrence throughout our lives) [125], lymphocytes with anti-self-reactivity, available in the periphery in healthy individuals, may be enough to kill cells early after their malignant transformation. However, this optimistic scenario changes radically when transformed cells escape to immune pressure and become less immunogenic. During this tumor development phase, severe impairment of intrathymic T cell differentiation/maturation has been reported, leading to a paralysis of cellular anti-tumor immunity [126,127,128,129,130]. Simultaneously, tumors produce large amounts of galectins (mainly galectins -1, -3, -7, -8, and -9). As already stated, increased levels of galectins in patients’ sera have been reported for several cancer types [27,28,29,30,31,32,33,34,35,36,37,38]. In such conditions of abundant production and secretion, galectins circulate through biological fluids and reach the thymus. Indeed, it was demonstrated that under stress conditions, high levels of circulating galectin-1 are associated with increased numbers of galectin-1-positive cells in the thymus medulla. This effect is not genetically mediated since no upregulation of galectin-1 mRNA was observed in thymic cells [131,132]. In this context, do galectins produced by the multiple cell types within a tumor impact T cell production and education in the thymus?

Since thymic education involves a high rate of cell death [133], galectins’ role in the regulation of thymic apoptosis awakes a particular interest. There exists information about the thymocyte pro-apoptotic role for most of the galectins produced by tumors (galectins-1, -3, -8, and -9).

Physiologically, galectin-1 is detected in thymic epithelial, endothelial, and dendritic cells, as well as macrophages [134,135]. From a functional point of view, galectin-1 induces thymocyte apoptosis [136]. This initial result was extensively confirmed under different experimental conditions, allowing a better understanding of this biological phenomenon. The highly proliferating immature CD4+ CD8+ double-positive thymocytes are the primary targets for galectin-1-induced cell death [136,137]. This FAS-independent apoptosis relies on a thymocyte permissive glycophenotype and involves the lectin interaction with CD7, CD43, and CD45 receptors [55,56,58,59,138]. Pro-apoptotic properties of galectin-1 on thymocytes have usually been evaluated in vitro using a soluble recombinant protein; only a few studies have used a more relevant biological context, demonstrating that galectin-1 produced in situ by cells have such a pro-apoptotic effect on T cell lines [46,139,140]. Furthermore, no report has addressed this issue using primary T cells. This point is crucial since galectin-1 is an inactive monomer that becomes a biologically active homodimer through non-covalent bonds with a Kd around 7 μM (equivalent to a concentration of 98 μg/mL) [141,142]. Based on this low homodimerization constant, high amounts of the protein are needed to reach the critical concentration required for the active dimer formation. Even if a genetically engineered dimeric galectin-1 reduces tenfold the concentrations of the protein to achieve a biologically active form [143], the required concentration of the lectin remains high.

Galectin-3 was also detected in epithelial and phagocytic cells in the medulla and, to a lesser extent, in the cortical regions of the thymus [74,144]. It is important to remark that galectin-3 has opposite effects on cells depending on its extracellular or intracellular localization [145,146]. Similar to galectin-1, extracellular galectin-3 induces thymocyte apoptosis. However, galectin-3 preferentially targets a different cell subpopulation (CD4- CD8- double-negative thymocytes) [74]. Furthermore, and although both galectin-1 and galectin-3 induced apoptosis are carbohydrate-dependent, galectin-3-mediated effects are different in several aspects: galectin-3 uses distinct sets of glyco-receptors (does not require CD7) and involves different molecular mechanisms [74]. Further, galectin-3 is more potent at inducing cell apoptosis when compared to galectin-1. However, galectin-3’s pro-apoptotic effects still require concentrations in the order of μM. In addition, this pro-apoptotic effect of extracellular galectin-3 opposes the anti-apoptotic function of intracellular galectin-3 [147]. Additionally, intracellular galectin-3 blocks galectin-1-mediated apoptosis [138], implying that both galectin members are closely interrelated in the control of thymocyte apoptosis.

Galectin-9 induces carbohydrate-dependent cell death in thymocytes [138]. Galectin-9 is detected in epithelial cells throughout the thymus, but it is more abundantly found in the medulla compared to the cortical regions of the thymus [138]. Again, galectin-9 has its particularities when compared against other galectins. Galectin-9 induces the cell death of all thymic subpopulations [138]; other galectins show more population-specific effects. Thymocytes’ apoptosis induced by galectin-9 involves receptors that are different from those used by galectins-1 and -3: while at present the relevant receptors remain unknown, CD44 could be a potential candidate since it has been demonstrated to bind galectin-9 in peripheral T cells [112,113]. At a mechanistic level, galectin-9-mediated apoptosis involves, at least partially, a Bcl-2-mediated pathway [138]. In addition, galectin-9 is more potent than the other galectins at inducing T cell death (<1 μM is effective) [138,148].

Galectin-8 is also found in the thymus but, in contrast to galectins-1, -3, and -9, it is not detected in thymic epithelial cells [149]. This galectin induces apoptosis of CD4+ CD8+ double-positive thymocytes through a mechanism that, at least partially, involves activation of the caspase-mediated pathway. In this in vitro study, concentrations of galectin-8 ranging from 0.5 to 2 μM were effective at inducing apoptosis [149].

Former evidence supports galectins acting as pro-apoptotic factors for thymocytes when produced in situ under physiological situations. Thus, galectins produced abundantly by tumors could shape the repertoire of newly generated T lymphocytes. As previously stated, galectins can circulate through biological fluids and reach the thymus. Although it is difficult to transfer in vitro concentrations to tissue levels, comparing the concentrations of circulating galectins in sera (in the order of ng/mL, as found in the 55 reports currently available for different cancers; some were cited before) with the concentrations of galectins required to trigger thymocyte apoptosis (in the order of μg/mL), the galectin concentrations reaching the thymus are likely insufficient to induce the thymocytes’ cell death. The only way tumor-derived galectins could induce thymocyte apoptosis would be by trapping these lectins, which would allow reaching the required galectin concentrations locally. To date, this phenomenon has not been described. Otherwise, if μM concentrations are reached in biological fluids, galectins may induce dangerous side effects, such as the aggregation of different types of cells [143,150] and potential systemic immunosuppression. Taking these arguments together, it seems unlikely that tumor-derived, circulating galectins can induce cell apoptosis in the thymus.

Apart from apoptosis, other biological properties, such as cell-to-cell interactions, can be regulated by galectins in the thymus [151]. For instance, galectin-3 was described as a factor promoting thymocytes’ release from thymic epithelial cells. Therefore this protein is a de-adhesive factor [144]. Conversely, a pro-adhesive role has been ascribed to galectin-1 through its interaction with several proteins of the extracellular matrix [134]. Thymic galectin-9 also acts as an adhesive molecule since it induces thymocyte homotypic aggregation [150]. Once again, all these biological aspects of galectins have essentially been addressed in vitro and require the use of high concentrations of recombinant proteins. It is, therefore, unlikely that any cell attachment-related modulation in the thymus can be ascribed to circulating and tumor-derived galectins.

Despite former arguments against the role of circulating and tumor-derived galectins in controlling thymic apoptosis and cell attachment-related functions, the evaluation of murine galectin-deficient models highlight that these proteins play essential roles in the control of thymocyte development. Indeed, while data identify endogenous galectin-1 as being not essential for thymocyte development (unaffected general thymic subpopulation numbers and percentages), this protein shapes the T cell repertoire acting as a selective modifier of positive and negative selection processes [152]. Indeed, physiological thymic galectin-1 opposes positive selection while it promotes negative selection of conventional CD8+ T cells. This conclusion was further confirmed by the use of a galectin-1-specific inhibitor in fetal thymic organ cultures; such treatment enhanced CD8+ T cell development [152]. The failures in thymic selection processes observed in galectin-1-deficient mice could explain (at least partially) some of the peripheral autoimmune phenomena spontaneously observed in these mice [153,154].

Endogenous galectin-3 is another member with an important modulation role in thymocyte development. Indeed, thymocyte subpopulations in galectin-3-deficient mice show a reduction in thymus cellularity, which is due to reduced in vivo proliferation and increased apoptosis of thymocytes [155]. Interestingly, although exogenous galectin-3 induces T cell apoptosis, the absence of endogenous galectin-3 also induces thymocyte apoptosis. This last point was demonstrated by genetic ablation and confirmed by specific inhibition experiments. Therefore, it is evident that optimal production of T lymphocytes requires physiological levels of galectin-3.

Up to mid-2021, no report indicates alterations in the thymic selection processes due to gene invalidation of galectins-7, -8, and -9. Nevertheless, it is interesting to note that galectin-7 is detected in Hassal corpuscules located in the medulla of the thymus [156]. While a functional significance remains unclear, this observation indicates a potential role of galectin-7 in thymic regulation. Altogether, data on galectins-1 and -3 supports that serum concentrations of galectins detected in cancer patients could modify the repertoire of T lymphocytes exported from the thymus to the periphery. Mechanisms by which this happens remain to be determined and more studies on the role of other galectins are necessary. Here, it is also important to note that, in addition to their lectin functions that require high concentrations to induce glycan–lectin lattices, galectins also act as regulators of gene expression, transcript maturation, and intracellular signaling [4,68,83,91,93,94,96,157,158,159,160,161]. Most of these later functions are glycan-independent and therefore require lower galectin levels [161]. The galectin concentrations detected in the circulation of cancer patients are consistent with these non-lectin functions. Therefore, galectins can act like soluble factors controlling the thymocyte educative process through effects on the thymocytes themselves or, indirectly, via effects on the stromal cells that participate in their education. Additional basic research can shed some light on these concerns.

### 1.2. T Lymphocyte Regulation by Galectins at the Periphery

In humans, the peripheral T cell pool is constituted by approximately 4 × 10^11^ naïve T lymphocytes [123]. In general, analysis of the murine peripheral T cell repertoire has demonstrated that each naïve clonotype is made up of few cells [162]. Similar low frequencies for each T clonotype have been found in humans [163]. More stringent restrictions apply to the anti-tumor-specific T cell repertoire since potent central selection processes only allow the exit of low avidity clonotypes against self-epitopes [164]. On the other hand, the frequency of anti-tumor T cells is increased through the clonotypes reactive against the neoepitopes, which arise as a consequence of tumor genetic instability; these clonotypes are not eliminated by central tolerance [165,166]. In general, basic studies have reported variable, low frequencies of pre-existing anti-tumor T lymphocytes in healthy individuals [167,168,169]. These low numbers of cells constitute the available army to recognize and eliminate transformed cells. In cancer patients, the said T cells expand and gain cytotoxic function in a tumor-dependent manner since each of these cancers differs in their mutational ratio, in their immunogenicity [170], and use of evasion pathways. Despite these considerations, the low number of specific anti-tumor T cells represents a real challenge for immuno-oncology. This fact implies that the expansion and gain of function of the few tumor-specific T cells must be as efficient as possible, at the risk of losing these effector cells. Hence, understanding the tumoral strategies leading to lymphocyte de-functionalization is essential to counteract them during cancer immunotherapies.

To assess the role of tumor-derived galectins on the function of immune cells in the periphery, we will focus on two anatomical locations where tumors have a direct and significant influence: the draining lymph nodes and the tumor itself. It does not imply that circulating galectins (produced by tumors) can influence cellular functions in other anatomical locations; these are beyond the scope of this review.

#### 1.2.1. Galectins’ Functions in Tumor-Draining Lymph Nodes

The first stages of lymphocyte activation occur in the draining node. There, a specific clonal expansion process is carried out [171]. Tumor-derived galectins can reach this anatomical location via blood and lymphatic vessels as soluble proteins, transported by cells or contained in exoparticles [172]. Once in lymph nodes, galectins impact on the early lymphocyte activation process. As mentioned above, there is abundant literature about the pro-apoptotic functions on recently activated T lymphocytes for the 5 galectins evaluated in this review. However, the concentrations of galectins found in blood and lymphatic fluids are unlikely to induce the glycolattice formation required for the said function. As previously discussed in the thymus section, it is therefore unlikely that the tumor-derived galectins in circulation modulate lymphocyte function in lymph nodes through their lectin properties. In contrast, other lymphocyte functions may be finely regulated by circulating tumor-derived galectins.

The emergence of a tumor is associated with a complete reorganization of the local tissue architecture, with major impacts on blood and lymphatic vessels. Indeed, circulating galectin-1 can be taken up by and control the functional properties of endothelial cells [20]. It is worth noting that galectin-1 expressed by endothelial cells plays a major regulatory role in the homing of naïve lymphocytes towards the lymph nodes [135,173,174]. Indeed, lymphocyte recruitment is significantly reduced in vitro. This phenomenon happens when endothelial cells are treated with recombinant galectin-1 at nM concentrations [173] or when endothelial cells upregulate galectin-1 following their incubation with tumor cell-conditioned media [135]. The latter is an example that mimics how tumor products can alter physiology, even at a distance. It is important to note that these nM concentrations (around 14 ng/mL) are compatible with the levels of galectins detected in biological fluids. More importantly, lymphocyte homing is significantly increased in galectin-1 deficient compared to wild-type mice [173]. This biological effect occurs independently of cell death [135] and both in physiological and inflammatory conditions [173]. Thus, tumor-derived galectin-1 decreases the influx of naïve T cells into the draining lymph nodes, accounting for a reduction in T cell activation and clonal expansion.

Despite regulation of cell migration through the blood endothelium, galectin-1 also plays a significant role in the formation of new lymphatic vessels. Indeed, genome-wide functional analysis revealed that galectin-1 is one of the major regulators of lymphatic endothelial cell function [175]. Therefore, this protein has a major impact on how tumor-derived antigens and antigen-presenting cells arrived at the draining lymph node through lymphatic vessels. Furthermore, galectin-1 inhibits the migration of immunogenic dendritic cells through the extracellular matrix and across lymphatic endothelial cells [176].

Galectin-1 compromises cell migration, and the T lymphocytes that effectively reach the draining lymph node are poorly activated if this lectin is present in the local media. Indeed, galectin-1 imparts a regulatory program in dendritic cells, resulting in lower lymphocyte priming [177,178]. However, regulation of the dendritic cell properties by galectin-1 does not exclusively depend on the extracellular concentrations of this lectin since endogenous galectin-1 also controls dendritic cell immunogenic potential [177,179,180]. Altogether, galectin-1 plays an essential role in controlling the initial steps of antigen-specific lymphocyte activation. Indeed, lymphocytes from the draining lymph nodes of galectin-1-silenced tumors are more prone to proliferation and produce higher levels of IL-2 and IFNγ [181,182]. The tumor origin of galectin-1 that causes alterations in antigen presentation is further supported by the fact that these biological effects are observed in the draining but not in other tumor-distant lymph nodes [181].

However, comprehension of the scenario in its full complexity requires additional clues. Indeed, the use of antigen-presenting cell-free systems demonstrated that galectin-1 modulates TCR-mediated signaling [137,183]. Accordingly, galectin-1 directly affects T cells during the early steps of activation, which are not only dependent on accessory cells. Altogether, tumor-derived galectin-1 promotes lymphocyte differentiation towards Th profiles that are inefficient to eliminate transformed cells [28,52,184,185,186]. In this respect, endogenous lymphocyte galectin-1 can control cell function at the level of gene expression regulation (reviewed in [4]). Indeed, endogenous galectin-1 in lymphocytes controls their expansion [187] and differentiation [188,189] in a variety of experimental models. In cancer, our group demonstrated that the inactivation of endogenous galectin-1 in lymphocytes reverses tumor immunosuppression [190].

Finally, tumor-derived galectins participate in the recruitment of cells with regulatory function in lymph nodes and thus have a major impact on the clonal expansion of anti-tumor T lymphocytes. Indeed, galectin-1 silencing in tumors reduces the frequency and the suppressive function of CD4+ CD25+ FOXP3+ regulatory T cells (Tregs) in draining lymph nodes [191]. Furthermore, Tregs require galectin-1 to be fully suppressive; galectin-1 neutralization reverses immunosuppression by Tregs [62,192]. Galectin-1 plays also an important role in the differentiation and suppressive function of CD122+ PD-1+ CD8+ Tregs [193]. In addition, this galectin also attracts other regulatory cells, such as M2 macrophages and myeloid-derived suppressor cells, to the tumor-draining lymph nodes, as it does towards the tumors themselves [194,195,196,197,198]. 

Galectin-3 is another member of this lectin family with a significant impact on the anti-tumor lymphocyte activation occurring in the draining lymph nodes. Our laboratory recently demonstrated that tumor galectin-3 is a potent negative checkpoint that suppresses lymphocyte proliferation in a prostate cancer microenvironment [199]. Furthermore, galectin-3 downregulation is a pre-requisite for optimal lymphocyte activation when a dendritic cell-based vaccine is used in prostate cancer. In such a case, long-term protective immunity is achieved [199]. Additional evidence suggests that galectin-3 could also act as a negative immune checkpoint in other types of cancers [72,75,79,200].

Among the molecular mechanisms accounting for the powerful lymphocyte inhibitory effect of galectin-3 in cancers, this protein modulates the interactions between T cells and antigen-presenting cells [85]. First, galectin-3 deficient immature dendritic cells have defective motility properties [201]. Consequently, by controlling the dendritic cell migration from the peripheral tissues (including tumors) to the draining lymph nodes, galectin-3 has a direct role in eliciting anti-tumor immune responses. Furthermore, this particular galectin also contributes to dendritic cell homeostasis since it was observed that galectin-3-deficient mice have increased numbers of plasmacytoid dendritic cells [79]. Interestingly, plasmacytoid dendritic cells are superior to conventional ones in activating anti-tumor CD8+ T lymphocytes [79]. Finally, information obtained from experimental models of infection has demonstrated the critical function of galectin-3 on the adaptive immune responses triggered by dendritic cells [202,203,204]. Altogether, these data seem to indicate galectin-3 plays a role at the initial steps of tumor antigen presentation.

Galectin-3 also has a direct effect on T lymphocytes. First, the galectin-3 expression on tumor cells negatively impacts the T lymphocyte numbers in lymph nodes [199]. This effect can be explained through the regulation at the initial steps of the lymphocyte activation process. Indeed, galectin-3 modulates the immunological synapse formation, restricting TCR movements, potentiating TCR downregulation, suppressing early TCR signaling pathways, and controlling cytokine production [73,76,85,205]. Extracellular galectin-3 accomplishes these biological effects via interactions with membrane glyco-receptors as well as by reducing the availability of soluble proteins (in particular cytokines like IL-2, IFNγ, and IL-12) [72]. These functions are glycan-dependent. Subject to the surrounding microenvironment, endogenous galectin-3 is upregulated in T cells early upon activation and skews their differentiation program. Indeed, galectin-3 deficiency promotes immune responses that favor effectors and effector memory T cells to the detriment of the generation of central memory T cells [85,206]. Furthermore, it was described that the survival of recently activated T cells might be affected by galectin-3. Indeed, in vitro studies have demonstrated that extracellular galectin-3 induces apoptosis in human T cells by directly binding the glycoprotein receptors CD45 and CD71 [74]. However, similarly to what was discussed on galectin-1, it is unlikely that tumor-derived circulating galectin-3 reaches the concentrations required to reveal its pro-apoptotic properties in the tumor-draining lymph nodes in cancer patients. It is also important to note that during activation, lymphocytes upregulate intracellular galectin-3 [206,207], which protects T cells from apoptosis [147,206]. Thus, the role of galectin-3 on the survival of newly activated lymphocytes in the lymph nodes is complex, and its real pathologic relevance remains controversial. On the other hand, the expression of galectin-3 by the stroma is required to recruit CD4+ CD25+ FOXP3+ Tregs towards immune organs in tumor-harboring mice [200]. Considering all these arguments, tumor-derived galectin-3 may substantially impact T cell activation, expansion, and polarization of the immune responses elicited in tumor-draining lymph nodes. This concept is relevant not only to the design of in vivo vaccine strategies [199] but also to adoptive T cell transfer of ex vivo-expanded tumor-reactive T cells [208].

Currently, few studies have evaluated the effect of galectin-8 in the process of anti-tumor immune activation. First, galectin-8 crosstalk among the VEFG-C, podoplanin, and integrin pathways plays a key role in lymphangiogenesis [209]. Indeed, podoplanin-expressing macrophages promote lymphangiogenesis in breast cancer via interaction with galectin-8 on lymphatic endothelial cells [210]. Some additional information can be drawn from other (non-tumoral) experimental models. Indeed, galectin-8 promotes all steps of antigen presentation from antigen binding, internalization, processing [211], and maturation of dendritic cells [212,213]. Studies with galectin-8 deficient antigen-presenting cells confirmed the relevance of such functions in pathophysiology [212]. This experimental model seems more relevant compared to artificial in vitro use of high concentrations of recombinant galectin-8. Aside from the antigen-presenting cell-dependent naïve CD4+ T cell co-stimulation that occurs with low galectin-8 concentrations, it was demonstrated that higher concentrations of galectin-8 induce antigen-independent proliferation of CD4+ T cells [214]. However, higher concentrations seem unlikely to be reached in tumor-draining lymph nodes, while low ones could play a role in controlling tumor antigen presentation in those immune organs.

Finally, recombinant galectin-8 increases differentiation of CTLA-4+ IL-10+ CD103+ Tregs through activation of TGF-β and sustained-IL-2 receptor signaling [215]. Tumors could use this strategy to block immunity in draining lymph nodes. In summary, little is known about the biological functions of galectin-8 in lymph nodes during cancer. Compilation of existing data indicates that secretion of this protein would not generate a strong selective advantage for tumors. On the contrary, galectin-8-based strategies could potentiate anti-tumor immunity since this lectin can lower the TCR activation threshold [216].

While the anti-tumor effect of galectin-9 was demonstrated in several experimental models [111,217,218,219], other strategies demonstrated galectin-9 has apparent diametrical opposite roles in immune regulation [220,221,222,223,224]. Possible explanations for these dichotomic observations range from the doses of galectin-9 used in the experiments, alternative regulations in different cell types, different receptors this lectin binds, different galectin-9 topographic localization in these cells (interaction with glyco-receptors exposed at the extracellular space contrasting to intracellular effects), different phases of the disease, and the activation of different signaling pathways in inflammatory versus tolerogenic microenvironments.

Interestingly, it was reported that low galectin-9 doses serve as an activation signal for resting T cells in the absence of any antigen stimulation. Indeed, exogenous galectin-9 initially causes the death of some naïve cells, but it induces robust proliferation of the surviving T cells [225]. Furthermore, exogenous galectin-9 increases early calcium mobilization and sensitizes T cells for higher IL-2 and IFNγ production [226,227]. These effects are antigen-independent and result in important changes in T cell phenotypes [225]. Therefore, tumor secreted galectin-9 can shape the available T cell repertoire by a direct action on resting T cells without any antigen stimulation.

During activation of anti-tumor-specific responses, exogenous galectin-9 modulates antigen presentation, promoting the differentiation of plasmacytoid dendritic-like cells [217], dendritic cell maturation, and a Th1 polarizing microenvironment [111,228]. Furthermore, the exogenous levels of galectin-9 serve as a potent regulator of cytokine production by T cells. This effect depends on the interactions with glycans but does not involve the T cell immunoglobulin and mucin domain-containing protein 3 (Tim-3) receptor [227]; this last being one of the main receptors described for galectin-9. On the other hand, the intrinsic expression of galectin-9 in T cells regulates early events of T cell activation. Indeed, galectin-9-deficient T cells show impaired proliferation, which cannot be recovered by exogenously added galectin-9 [229]. It was demonstrated that this galectin is recruited to the immune synapse upon T cell activation, contributing to proximal TCR signaling [229]. Altogether, these results indicate a positive regulation of T cell activation and expansion by galectin-9.

Galectin-9 also promotes immune regulation. During the initial events of naïve CD4+ T cell activation and clonal expansion occurring in lymph nodes, some T helper cells secrete galectin-9 with the concomitant production of IL-10 and TGFβ [230,231]. In this cytokine scenario, conventional T cells acquire regulatory functions (so-called induced regulatory T cells, iTreg). Galectin-9 is, therefore, a paracrine and autocrine factor that enhances the expansion and suppressive phenotype of iTregs [113]. Indeed, galectin-9 interacts with the CD44–TGF–βRI complex to stabilize the expression of FOXP3, the master transcription factor of conventional Tregs. Besides controlling CD4+ CD25+ FOXP3+ Tregs function, galectin-9 also clusters the 4-1BB receptors, promoting in vivo accumulation of CD8+ Tregs, an alternative suppressive pathway activated during inflammation [115]. Furthermore, galectin-9 binds the V-domain Ig suppressor of T cell activation protein (VISTA). This interaction induces T cells to acquire a dysfunctional phenotype [232]. Galectin-9 also promotes CD11b+ Ly-6G+ myeloid-derived suppressor cells, an additional point of regulation of T cells expansion. This effect on myeloid suppressor cells requires the interaction of galectin-9 with the Tim-3 receptor [233]. However, several inducible receptors in T cells are involved in galectin-9-mediated immunoregulation: Tim-3 [234], CD44 [113], DR3 [117], 4-1BB [115], CD40 [116], and the protein disulfide isomerase [118].

In addition, galectin-9 is also described as an apoptosis inducer of activated T cells (discussed in depth in the next chapter). However, the concentrations of lectin required to induce cell death (in the order of nM) are unlikely to be reached in the tumor-draining lymph nodes, precluding any pro-apoptotic role of tumor-derived galectin-9 at this anatomical localization. Finally, it is important to note that galectin-9 interventions in cancer may take some considerations on kinetics and receptor expressions into account. For instance, TCR downregulation is a well-known phenomenon occurring early during T cell activation in lymph nodes [235]. Considering that some of the described functions for galectin-9 require the integrity of the TCR/CD3 signaling pathway (such as calcium mobilization) [226], these functions of galectin-9 may not be relevant in recently activated lymphocytes in tumor-draining lymph nodes.

#### 1.2.2. Galectin Functions inside the Tumor

Tumor T cell infiltration and effector functions are important biomarkers for predicting better clinical outcomes [236,237,238]. Several preclinical models have evaluated the impact of galectins-1, -3, -7, -8, and -9 produced by tumors in controlling T cell behavior. It is important to note that tumors are made up of different types of cells, including transformed cells and non-transformed stroma (fibroblasts, macrophages, endothelial and immune cells, among others). All of these cells contribute to the tumor production of galectins.

In the case of galectin-1, experimental evidence demonstrated that galectin-1 from the tumor, and not from the host, plays a fundamental role in contributing to tumor growth and distant metastasis [194,239,240]. Consequently, experiences in which galectin-1 was inhibited in tumor cells have shown that this lectin serves as a potent pro-tumor agent [174,181,182,194,239,240]. The mechanisms by which this regulation happens remain a matter of discussion. However, it is incontestable that galectin-1 pro-tumor effects require the active participation of the immune system. Indeed, tumors expressing or not galectin-1 grow indistinctly in immunodeficient mice [52,181,239,241,242], clearly indicating that the immune system is the major target of tumor-galectin-1. Furthermore, immune cell depletion experiences indicate that CD4 and CD8 T lymphocytes are involved in the effects of galectin-1 [181,194]. This concept is also supported by the use of CD8 T lymphocyte-deficient mice [174]. Additionally, cells of innate immunity could also play a direct or indirect role in these biological phenomena [194,197,243].

What are the mechanisms by which tumor-derived galectins accomplish immune deactivation? A hypothesis raised by Van der Brûle in 2001 supports the concept of galectin-1 serving as a tumor-protective shield since this lectin induces the death of effector cells reaching the tumor [39]. Indeed, a seminal article published in 1995 was the first to demonstrate that galectin-1 induces apoptosis of T lymphocytes [139]. While this pioneering result showed that all T lymphocytes bind galectin-1 [139], further studies have revealed that this protein has the opposite biological effects in naïve versus activated T cells. While galectin-1 (at low concentrations) promotes survival of resting naïve T cells [244], activated Th1 effector cells have a permissive glycophenotype that allows galectin-1 to induce their apoptosis [139,185,245]. The latter include effector cells that should migrate toward tumors to accomplish their cytotoxic function.

However, the pro-apoptotic effects of galectin-1 on activated T cells remain controversial in several aspects. First, regarding in vitro methodological issues related to the use of reducing agents in galectin-1 preparations, the said reagents per se can induce apoptosis in T cells [246]. Therefore, in vitro studies that use recombinant galectin-1 obtained in such conditions should be analyzed with caution. Furthermore, the evaluation of the entire in vivo picture highlights that even if tumors produce abundant galectin-1, their microenvironment is highly oxidative (high levels of oxygen and nitrogen derived-species resulting from its metabolism) [247]. The oxidative microenvironment together with the requirement of high protein concentrations for homodimerization may make it difficult for galectin-1 to reach optimal conditions to induce apoptosis of the infiltrating effector lymphocytes. Therefore, this tumor pro-apoptotic function seems to require particular conditions: galectin-1 reaching locally high concentrations and protected from the oxidative microenvironment. In this sense, after secretion, galectin-1 can remain within the cell membrane and be associated with components of the extracellular matrix. At this localization, tumor galectin-1 is more effective in its pro-apoptotic activity [46]. This phenomenon is crucial for preserving its lectin activity since the binding of galectin-1 to sugar residues on membrane proteins prevents it from being oxidized by the cell environment [141,248]. In addition, galectins can also compartmentalize in particular cell membrane structures (e.g., lipid rafts where some galectins have been detected [201,249]) and therefore be protected from the extracellular redox damage [118]. These observations indicate that tumors can induce topographic sites of intimate cell-to-cell contacts that privilege galectin-1 expression in an active and T cell pro-apoptotic form. This concept is supported by very few reports that have evaluated the in vivo tumor lymphocyte pro-apoptotic effect [239] or by using relevant co-culture systems [186,250,251,252]. This information also indicates important differences between the galectin-1 effects in the tumor and other lymphoid organs. Indeed, at the tumor site, galectin-1 may exert its lymphocyte pro-apoptotic effect only if the microenvironment allows a close contact between lymphocytes and tumor cells. In contrast, the galectins produced by the tumor could not be considered as soluble factors with T cell pro-apoptotic functions in other lymphoid organs (as developed in previous chapters).

Nevertheless, other experimental models do not support the in vivo lymphocyte pro-apoptotic effects of tumor galectin-1 [174,253]. Therefore, alternative mechanisms should also be considered. While the growth of wild-type tumor cells is hampered in galectin-1-deficient mice, the same tumor cells are tumorigenic in wild-type mice [20,190,241,243]. This observation supports the relevance of the tumor-associated stroma in galectin-1 mediated effects. On the one hand, galectin-1 is a potent pro-angiogenic factor [20,25,51,52,254,255]. Besides providing tumors with nutrients/oxygen/hormones, etc., the formation of new blood vessels should facilitate the arrival of lymphocytes and promotes tumor infiltration (if the correct adhesion molecules are expressed). This is not the case in several models; the expression of galectin-1 in stromal endothelial cells induces less lymphocytic infiltration [135,256]. Furthermore, the galectin-1 produced by tumors is taken up by endothelial cells, activates these cells, and promotes neovascularization [20,257,258], but inhibits lymphocyte migration through the endothelium and the extracellular matrix [135,173,174]. The net result is lower intra-tumoral lymphocyte infiltration [52,182,197,253,256]. In this case, the concentrations of galectin-1 detected in pathophysiological situations (around nM) are capable of reducing lymphocyte attachment and adhesion to activated endothelium [173,174]. Furthermore, it was also nicely demonstrated that tumor-derived galectin-1 upregulates PD-L1 and galectin-9 in endothelial cells. This molecular pathway is responsible for the inhibition of lymphocyte migration towards tumors, reversing the immune checkpoint resistance observed in some cancer patients [174]. Thus, a concept that deserves increasing attention positions galectin-1 secreted by tumors as a paracrine molecule acting on the stroma to create a microenvironment permissive for tumor growth. In this context, our group demonstrated that tumors significantly impact the intracellular levels of galectin-1 in T lymphocytes. This phenomenon is a powerful way by which tumors control T cell proliferation and effector properties. Indeed, lymphocytes with higher levels of galectin-1 show lower cytotoxic effector capacity within a tumor contexture [190]. Experimental data also support that the intrinsic galectin-1 in T lymphocytes have major regulatory functions in other models of T cell immunity: infections, allergic contact dermatitis, and autoimmunity [188,189,259,260]. Aside from direct regulation of T cell function, galectin-1 may also control lymphocyte behavior through the participation of other cells. Indeed, several types of tumors are characterized by significant Treg cell infiltrates (reviewed in [261]). This observation indicates that suppression of T cell expansion does not only occur at the draining lymph nodes. Tumors can be considered tertiary immune organs where T cell proliferation is still possible and thus must be suppressed. In this context, the galectin-1 produced by tumors may control the function of Tregs in a paracrine manner [191,192,262]. Indeed, blockade of exposed galectin-1 in Tregs reduced their suppressive properties [192]. This regulatory effect of galectin-1 is such a potent mechanism that it can also control immunity at distant metastatic sites [191]. Apart from Tregs, other immune-suppressive cells are also modulated by tumor-derived galectin-1. Indeed, intra-tumor accumulation of myeloid cells depends on tumor-derived galectin-1 [194,197,263], a phenomenon intimately related to the microbiota through the participation of γδ T cells [243]. This last observation evidences the complex cellular network that coordinately regulates immune responses against tumors, some occurring at a distance from the tumor site.

Tumor galectin-3 is an additional potent regulator of lymphocyte effector properties at the tumor site [72,199]. First, T cell recruitment towards the tumor requires an IFNγ-induced CXCL9/10 gradient of chemokines. It has been demonstrated that IFNγ entrapment by galectin-3 in the tumor extracellular matrix prevents the creation of such a chemokine gradient required to attract T cells towards the tumor [72]. The relevance of this phenomenon was also confirmed after demonstrating that galectin-3 interacts with glycans of IFNγ-R2, therefore interfering with Janus-activated tyrosine kinase/signal transducer and activator of transcription (JAK/STAT) activation (an essential signaling pathway in responding T cells) [264]. Besides regulating T cell migration, galectin-3 also controls effector functions. Indeed, galectin-3 silenced tumors elicit lymphocytes with increased degranulation capability and specific anti-tumor cytotoxicity [199]. Mechanistically, it was demonstrated that galectin-3 decreases lymphocyte effector functions through downregulation of the TCR, thus transforming T infiltrating lymphocytes (TILs) into dysfunctional cells [75]. These defective rearrangements at the T cell synapse induced by galectin-3 are responsible for the failure of CD8+ TILs to attach to their specific targets and to secrete killing cytokines (even while these cells produce normal amounts of cytokines) [76]. These galectin-3-mediated effects are local (inside the tumor) since the functional unresponsiveness of TILs contrasts with the functional properties of blood T cells [75,265]. The dysfunctional state of intra-tumor lymphocytes induced by galectin-3 is further supported by their expression of terminal T cell deactivation/exhaustion markers [266]. Indeed, the CD8+ TILs that bind galectin-3 in the tumor microenvironment co-express LAG-3 and PD-1 [79]. This study demonstrated the central role LAG-3 plays in galectin-3-mediated suppression of lymphocyte anti-tumor effector function [79]. All these arguments support the significant role of galectin-3 in reducing tumor infiltration and the killing function of activated T lymphocytes.

In addition, and since galectin-3 was described as pro-apoptotic for activated lymphocytes [74,145,147,267], the same considerations previously stated for galectin-1 apply for galectin-3. Indeed, galectin-3 interacts with proteins in the tumor extracellular matrix and colocalizes in the cell lipid nanodomains [201,268]. Thus, galectin-3 reaches high protein levels and is protected from the oxidative microenvironment.

Apart from a direct effect on effector T cells, galectin-3 attracts macrophages towards the tumor and promotes their M2 differentiation [269]. Furthermore, other innate immune cells may also be involved in the complete picture of how galectin-3 regulates immunity against cancer [200,205]. Altogether, these arguments clearly state that galectin-3 plays a central role in tumors escaping the anti-tumor effector function. This concept must be taken into consideration in the immunotherapy approaches. Indeed, we recently demonstrated that preconditioning the tumor microenvironment through galectin-3 downmodulation is a pre-requirement for a long-term protective vaccine strategy in prostate cancer [199].

Galectin-7 is detected in several types of tumors [270,271,272,273,274,275,276,277]. In general, higher histological galectin-7 expression in tumors represents a negative prognostic factor for the survival of cancer patients [274,275,276]. A transcriptome meta-analysis of cervical cancer cells after ectopic galectin-7 expression demonstrated the regulation of molecular networks, such as metabolism, growth control, invasion, apoptosis, and control of the immune response [278]. This result suggests that galectin-7 plays a role in the tumor microenvironment and immune surveillance. Interestingly, galectin-7 is detected in tumor-associated macrophages [274]. Whether macrophages produce galectin-7 themselves or take it up from neighbor tumor cells remains unknown [22,279], nor is the functional impact of galectin-7 on tumor-associated macrophages. On the other hand, recombinant galectin-7 induces apoptosis of human peripheral T cells [280]. As previously mentioned, the biological concentrations required to induce active galectin isoforms, its topographic cellular distribution, and the oxidative properties of the surrounding environment are important parameters in determining the pro-apoptotic function of this lectin. Apart from controlling the survival of activated T cells, galectin-7 could also be used by tumors to downregulate the cytokine production by T cells, as was demonstrated in other experimental models [281]. Altogether, galectin-7 plays interesting effects on the function and survival of immune cells and should be considered a target for immunotherapies.

Currently, there are no studies evaluating the impact of tumor galectin-8 on the generation of specific and protective immune responses. Most studies have focused on the role of this protein in tumor metastasis and angiogenesis [103,282,283,284,285]. However, inspiration from non-tumoral scenarios can illustrate some potential immune functions of this protein at the tumor site. Galectin-8 promotes cell death of activated lymphocytes. Indeed, Phytohemagglutinin (PHA)- and CD3/CD28-pre-stimulated human peripheral blood mononuclear cells (PBMCs) become apoptotic after re-stimulation in the presence of recombinant galectin-8 [286,287]. Similarly, galectin-8 provides anti-proliferative signals to pre-activated T cells [286,287]. Together with those data exposed in the lymph node chapter, these results indicate a dual role of galectin-8 in T cell function. First, galectin-8 enhances (co-stimulates) early T responses in the lymph nodes, especially when the stimulus is limited or when the available T cell repertoire has low avidity for epitopes (cancer is such a case). Second, galectin-8 restricts the effector phase of ongoing immune responses through a direct pro-apoptotic effect. However, more experimental data are needed to clarify these biological effects of galectin-8 in cancer.

Contradictory results were observed in several studies that evaluated the prognostic value of the galectin-9 expression in solid tumors. Galectin-9 is a positive prognosis biomarker for patients with some types of cancer [288]. This positive correlation between galectin-9 expression and the overall survival can be accounted by direct effects on tumor cells such as inhibition of metastasis [289,290] and apoptosis induction [291], and also indirectly, through the induction of a more efficient anti-tumor immune response in the draining lymph nodes (as stated in the previous chapter). However, as tumor settles as a chronic disease, galectin-9 expression correlates with a poor prognostic value and often associates with immune evasion [110]. Intratumor CD103+ dendritic cells express Tim-3; its interaction with galectin-9 induces deactivation of antigen-presenting cells [292]. Interestingly, this subset of dendritic cells is superior in antigen transport from tumors and its cross-presentation in draining lymph nodes [293,294,295]. In addition, this dendritic cell subtype plays a fundamental role in the local re-stimulation of CD8+ T cells [296]. Furthermore, tumor galectin-9 interacts with dectin-1 on macrophages to promote their tolerogenic program [297]. Therefore, tumor galectin-9 plays a significant role in controlling myeloid cell’s properties, T cell activation and also in controlling the effector phase of anti-tumor immune responses.

After being activated in the lymph nodes, the migration of lymphocytes into tumors is regulated by galectins. Indeed, tumor-derived galectin-1 remodels the local endothelium in a way that galectin-9 and PD-L1 are upregulated to prevent the lymphocyte infiltration of tumors [174]. Furthermore, galectin-9 contained in tumor-derived exosomes [172] attracts regulatory T cells towards tumors [298]. As a consequence of their high levels of Tim-3 [299,300], Tregs are very sensitive to the galectin-9 levels in the tumor microenvironment, and this galectin-9/Tim-3 signaling pathway plays a major role in dampening any potentially lymph node-elicited, anti-tumor immune response.

Not only do tumor cells express galectin-9, but this protein is also detected in tumor-infiltrating Tregs [113,234] and tumor-associated macrophages [301]. This particular cellular microenvironment induces arriving T cells to acquire a characteristic PD1+ Tim3+ CD8+ “exhausted” phenotype, which is associated with failure of T cell proliferation and effector function [301,302,303]. In addition, it was demonstrated that through interaction with Tim-3, galectin-9 induces apoptosis of effector Th1 lymphocytes [109]. Since galectin-9-mediated cell death is not completely abolished in Tim-3-deficient cells, galectin-9 may also use additional receptors to induce Th1 cell death [109,227]. Indeed, tumor galectin-9 also interacts with VISTA in T lymphocytes [114]. This interaction results in the activation of granzyme B inside cytotoxic T cells, causing their apoptosis [114]. Regardless of the mechanism, CD4+ and CD8+ T lymphocytes are both sensitive to galectin-9-mediated apoptosis [148,220]. It must be noted that, compared to the other galectin members, lower concentrations of galectin-9 (in the order of nM) are required to induce T cell apoptosis [226]. Galectin-9/Tim-3 interactions occur in lipid rafts [304]; these types of interactions have well-defined topographic locations. In such membrane domains, galectin-9 binds to and increases retention of the protein disulfide isomerase (PDI) at the cell surface, thus controlling the redox status at the plasma membrane [118]. Finally, galectin-9 induction of T cell apoptosis seems to be finely regulated. Indeed, when expressed at high levels, PD-1 also binds galectin-9 in a glycan-dependent manner, and the co-expression of PD-1 and Tim-3 protects T cells from galectin-9-induced apoptosis [110]. Through this mechanism, tumor-galectin-9 eliminates the effectors but not exhausted T cells. These results explain why dysfunctional PD-1+ Tim-3+ T cells persist within the tumor microenvironment and even dominate the intratumoral CD8+ T cell population in several cancers [305,306,307].

## 2. Conclusions and Future Directions

The low frequency of cancers that develop all along our lives attests to the high efficiency of the immune system to eliminate early transformed cells. Said efficiency occurs even if few immunogenic tumor antigens have been described, and very low numbers of tumor-specific T lymphocytes (mostly with low avidity TCR) are detected in a healthy individual [169,308]. Nevertheless, as the tumor is settled and progresses, it develops powerful immunosuppression mechanisms to escape immune attacks. In recent years, much progress has been made in understanding such regulatory mechanisms, the so-called negative checkpoints [309]. However, the partial clinical results obtained with such interventions indicate the need for a more comprehensive understanding of the entire process of anti-tumor immune activation [310].

This review highlights that tumor-derived circulating galectins can affect cellular and molecular processes in central and peripheral immune organs to prevent the immune attack of transformed cells. Most of the immunological functions that have been ascribed to galectins in cancer were analyzed in this review. Other functions are likely still worthy of discovery. Among those already known, one of the most studied functions of galectins concerns their ability to induce apoptosis of activated T lymphocytes [311]. A comprehensive evaluation of the existing experimental evidence indicates this effect would be limited to particular structures within the tumor itself. Therein, galectins would reach the necessary concentrations to be active and would be protected from the tumor oxidative environment [118]. On the contrary, biological functions that do not require the formation of lattices between oligomerized galectins and glycans exposed at the cell membrane and the extracellular matrix can be accomplished with lower concentrations of galectins. Therefore, such glycan-independent functions are more easily achievable at tumor-distant tissues (e.g., thymus, draining lymph nodes, and other immune organs in patients harboring a tumor). Among these functions, it is worth mentioning that circulating galectins can be taken up by immune cells [20,21,22,132]. Once they reach the intracellular space, they interact with cytoplasmic and nuclear molecules, resulting in the control of cell behavior, including cell gene expression (reviewed in [4]). While these non-lectin interactions have received less attention until now, their impact on cancer and other pathologies should not be undervalued. Thus, and without requiring high concentrations, galectins may function as soluble factors affecting each stage of the anti-tumor immune response (T cell migration/activation/effector function). Therefore, galectins represent attractive targets for intervention in cancer immunotherapy.

In this scenario, increasing experimental and clinical evidence indicates that galectins’ blockade as monotherapy does not result in any significant advantage for cancer treatment [42,110,196,292,312,313]. However, galectins are involved in patient sensitivity or resistance to chemo-, radio-, immune-, anti-angiogenic, and targeted-therapies (reviewed in [314]), promising that effective therapeutic avenues can be achieved by combining galectins’ inhibition with the former strategies. It should be noted that said combinatory strategies may involve processes that are immune-dependent as well as others that are not. Among the last ones, we can mention that inhibition of galectin-1 or galectin-3 both potentiates tumor cell sensitivity to several types of chemotherapies (involving different molecular mechanisms) in a panoply of different cancers [315,316,317,318,319,320,321,322,323,324,325,326,327,328,329,330,331,332,333,334,335]. On the other hand, administration of galectin-9 increases the sensitivity of chronic myeloid leukemia to the BCR-ABL tyrosine kinase drug imatinib [336]. Many of these described mechanisms support the combination between galectins’ inhibition and targeted therapies. In-depth evaluation of these galectins’ immune-independent functions is beyond the scope of this review. However, they must be carefully considered to define a personalized combinatorial therapeutic strategy for each patient.

Interestingly, the galectins’ inhibition combined with chemotherapy impacts the anti-tumor immunity (Table 1). In colorectal liver metastasis, single-cell analyses defined two mutually exclusive subsets of tumor cells with divergent response to chemotherapy: the stem-like cells (tumors cells which mainly use the PD-1/PD-L1 pathway to control immunity) and the enterocyte-like cells (which use the Tim-3/galectin-9 pathway to evade immunity) [337]. This observation highlights the impact of chemotherapies on the immune system’s ability to attack tumor cells and the need to select combinatorial strategies carefully. In breast cancer, Tim-3 positivity was associated with a worse chemotherapy response [338]. Besides, the use of neutralizing anti-Tim-3 or anti-galectin-9 antibodies improves paclitaxel-based chemotherapy [292]. In said cases, combinatorial treatments induce negative regulation of tumor growth by mechanisms that depend on CD103+ dendritic cells and CD8+ T lymphocytes. Upon such a combinatory treatment, CD103+ dendritic cells express higher levels of CXCL9 chemokine ligand, which attracts CD8+ T lymphocytes towards the tumor. Indeed, not only do increased numbers of CD8+ T cells infiltrate tumors, but these cells also have higher effector functions [292].

The combination of galectin-1 inhibition and chemotherapy is another promising strategy for some types of cancers. Indeed, synergic therapeutic effects were reported by combining inhibition of galectin-1 and temozolomide to treat glioblastoma [196]. Such combinatory treatment switches macrophages to M1 polarization, reduces myeloid-derived suppressor and regulatory T cells, and increases tumors’ CD4+ and CD8+ T cells infiltration [196].

Interestingly, a positive correlation between circulating galectin-3 levels and paclitaxel resistance was demonstrated in patients with ovarian cancer [339]. In those patients, paclitaxel triggers the TLR-4/MyD88 pathway signaling [340], and exogenous galectin-3 boosts such signaling and promotes higher levels of IL-6, IL-8, and VEGF release [339]. This observation further supports that exogenous galectin-3 plays immune-mediated roles during chemotherapies. In prostate cancer, low doses of docetaxel downregulate tumor galectin-3, even in docetaxel-resistant patients [199]. As a result of the said tumor galectin-3 inhibition, vaccination induces long-term anti-prostate cancer immune protection [199]. This observation highlights a molecular mechanism (mediated by galectins) explaining the synergy between chemotherapy and immunotherapy and the importance of the chronology between both treatments. While inhibition of galectin-3 before vaccination is efficient, all standard clinical assays using the opposite chronology seem not to benefit patients’ survival [42].

Galectin inhibition may also be a good strategy combined with radiotherapy. It was demonstrated that radiotherapy increases the tumor levels and secretion of galectin-1 [341,342]. High levels of circulating galectin-1 are directly associated with lymphopenia [342] and radioresistance [343] in cancer patients. Furthermore, increased galectin-1 levels resulting from radiation is one of the main causes of poor tumor T cell infiltration, the tumor endothelium being a major actor in this process [174,344]. Therefore, irradiation creates an unfavorable immune contexture to mount efficient anti-tumor immune responses. Besides, the galectin-1 blockade can help to reverse such adverse immune scenarios. Accordingly, the use of Anginex (a 33 amino acid galectin-1 inhibitory peptide) combined with a suboptimal dose of radiation causes tumor regression in ovarian, mammary, and squamous cell carcinoma models [313].

Tumor-induced hypoxia is a major driving force that promotes angiogenesis and impairs effector immune responses [345,346]. Indeed, in vitro studies have demonstrated that hypoxia promotes the differentiation of PD-1+ Tim-3+ terminally exhausted-like CD8+ T cells [347]. Furthermore, these exhausted T cells are resistant to inhibitory checkpoints strategies [348]. Altogether, these data indicate a close regulation between hypoxia, angiogenesis, and immunosuppression. Antiangiogenic agents have been shown to normalize tumor vasculature transiently, which alleviates hypoxia, improves the response to various chemotherapies, and facilitates immune cell infiltration of tumors [349]. In this context, inhibition of galectin-1, similar to other antiangiogenic agents, resulted in transient vessel normalization, as evidenced by vasculature remodeling, increased pericyte coverage of vessels, and T cell infiltration, as well as reduced tumor hypoxia [52,196,350]. Therefore, inhibition of galectins can provide a rational basis to optimize synergistic combinations of antiangiogenic and immunotherapeutic strategies, with the overarching goal of improving the efficacy of these treatments.

Several reports have demonstrated that hormone-sensitive cancers change their galectinome signature during disease progression [255,351,352]. Besides being interesting biomarkers for prognosis and treatment resistance, galectin inhibition in these cancers represents attractive strategies to be combined with hormone-related ones. Although androgen-deprivation therapy, the most common treatment for prostate cancer, initially promotes a robust T cell infiltrate, T cell responses are later attenuated due to potent tolerogenic mechanisms tumors develop [353,354]. Given the immunosuppressive functions described for galectins, their participation in such immune phenomena upon androgen deprivation seems plausible. On the other hand, progestagens reconfigure the breast cancer microenvironment, inducing the preferential expansion of myeloid-derived suppressor cells in the spleen and bone marrow of tumor-bearing mice [355]. Furthermore, anti-progesterone treatments enhance the anti-tumor immune response and increase sensitivity to the PD-L1 blockade in breast cancers [356]. It is worth noting that progesterone regulates galectin-1 expression in some experimental settings [357,358]. However, there is still a lack of direct evidence between galectins and hormone-dependent immune escape. Indeed, more research is needed to clarify if the galectins’ inhibition could have an additive effect on hormone-dependent therapies by preventing immune escape mechanisms induced by such approaches.

Galectin inhibition can also potentiate other immunotherapies. For instance, galectin-1 knockdown synergizes with dendritic cell vaccination and PD-1 blockade in glioblastoma [196]. Data from head and neck cancer demonstrated that galectin-1 inhibition enhances anti-PD1 therapy, suggesting that a combination of galectin-1 inhibitors and anti-PD1/PDL1 immune checkpoint synergize for cancer treatment [174]. Indeed, galectin-1 induces resistance to immunotherapy through upregulation of PD-L1 and galectin-9 in the endothelium, resulting in impaired T cell infiltration [174]. In addition, OTX008, a selective galectin-1 inhibitor, inhibits tumor growth in several preclinical studies [350,359,360]. In 2012, a phase I clinical trial aiming at evaluating the effects of subcutaneous administration of OTX008 for the treatment of advanced solid tumors was announced (ClinicalTrials.gov: NCT01724320, accessed on 16 July 2021). No results have been communicated up to the present. Although the effects of OTX008 on cancer-associated immune system remain misunderstood, some information can be drawn from leukemia. Indeed, inhibition of leukemia patients’ serum galectin-1 by OTX008 abrogates IL-10 production by dendritic and T cells [361], implying this molecule prevents immune deactivation. More studies are needed to fully understand the effects of OTX008 and other galectin-1 inhibitors on the immune system in cancer.

In lung cancer, the accumulation of Tim-3-expressing lymphoid cells and galectin-9-expressing monocytic myeloid-derived suppressor cells positively correlates with resistance to anti-PD1 immunotherapy [312]. Accordingly, resistance to anti-PD-1 can be overcome by in vitro blockade of the galectin-9/Tim-3 pathway [362]. This type of results supports a phase I/II trial that currently evaluates the safety and efficacy of MBG453, an antibody against Tim-3, as a single agent and in combination with anti-PD-1 in patients with advanced malignancies (ClinicalTrials.gov: NCT02608268; accessed on 16 July 2021).

Galectin-3 has been involved as a major determinant of cold tumors (those which do not respond to immune checkpoint inhibitors due to the paucity of tumor T cell infiltration, as happens in prostate and pancreas adenocarcinomas); galectin-3 inhibition can reverse such resistance [363]. Consequently, inhibition of galectin-3 in tumor cells leads to an optimal anti-prostate cancer vaccination [199]. In the same way, galectin-3 inhibition in a pancreatic adenocarcinoma allogeneic vaccine increased disease-free survival in patients [79]. Nevertheless, the impact of galectin-3 inhibition seems to apply to other cancers, including hot cancer. Oral administration of a galectin-3 inhibitor (GB1107) reduced human and mouse lung adenocarcinoma growth and blocked metastasis in a syngeneic model [364]. Treatment with GB1107 increased tumor M1 macrophage polarization and CD8+ T cell infiltration. Furthermore, GB1107 potentiated the effects of a PD-L1 immune checkpoint inhibitor to increase the production of cytotoxic effector molecules [364]. In metastatic melanoma and head and neck cancers, anti-PD-1 treatments are improved by belapectin (GR-MD-02), a galectin-3 inhibitor [365]. This combinatory therapy significantly increases effector memory T cell activation and reduces monocytic myeloid-derived suppressor cells detected in blood [365]. In non-small cell lung carcinoma, high galectin-3 expression correlated with a poor response to PD-1 blockade in a small cohort of patients [366]. Altogether, the available information underlines the key roles played by galectins in resistance to anti-PD-1 treatments and supports how useful galectin’s inhibition could be when associated with a negative checkpoint’s blockade. Accordingly, several clinical trials are in progress to treat patients with galectin-3 inhibitors and immunotherapy in different tumor types. These clinical trials include non-small cell lung cancer, squamous cell head and neck cancers (ClinicalTrials.gov: NCT02575404), and melanoma (ClinicalTrials.gov: NCT02117362, NCT02575404, both accessed on 16 July 2021) (see Table 1, [42]). Publication of such results will surely clarify the potential role of galectin inhibitors in treating those cancers.

Finally, it is interesting to note that cancer patients following different therapies show better clinical outcomes if they develop galectin-neutralizing antibodies, a natural way to prevent their biological functions [79,367,368,369]. This argument strongly supports the potential use of galectin inhibitors combined with other approaches to reach more effective treatments for cancer patients.

Even though several galectin inhibitors have been described for more than a decade (Table 2), none has had clinical success [42]. Several reasons may explain such disappointing results. First, most of these inhibitors display weak affinities for galectins [370,371,372,373,374]. Second, in general, the available molecules inhibit various galectin members. This point is challenging since different galectins display high sequence homology [375,376]. The development of member-specific inhibitors would be advantageous at reducing the minimal effective dose required to obtain functional results and prevent side effects. Several inhibitors also interact with galectins via their carbohydrate recognition domain (CRD) [377]. In general, such inhibitors are not effective at targeting the CRD-independent functions of galectins [370]. Moreover, since galectins display intra- and extracellular functions, the permeability of these molecules is challenging for their biological effects. Molecular weight is another relevant biochemical parameter to take into account when polysaccharide-derived or polymeric inhibitors are used. Both parameters determine the biodistribution of these molecules, defining which tissues and cells they can access. This review attempts to demonstrate the relevance of these inhibitors reaching the primary and secondary lymphoid organs and the tumor itself. In addition, most glycan-mimicking inhibitors are sensitive to enzymatic hydrolysis by glycosidases, reducing the kinetics of their biological effects [378]. Nevertheless, since galectins are involved in physiological processes, their inhibition can potentially induce side effects. Therefore, it would be desirable to inhibit galectins only at the right place, preventing the systemic biodistribution of these molecules. Considering the abundant experimental evidence supporting galectins as major intermediaries in tumor immune escape, it would be interesting to develop strategies to block them locally in immune-related tissues. As discussed in this review, lymph nodes (where anti-tumor immune responses are elicited) and the tumor itself (where effector function is performed) are attractive anatomic sites where galectins may be inhibited. Finally, it was demonstrated that tumors generate different isoforms of galectins via alternative splicing [379]. This phenomenon can induce inhibitor-resistant galectins. Combinatory strategies are, therefore, logical strategies to prevent resistance selection.

In conclusion, to block galectin-mediated tumor immune escape, the scientific community needs to develop more effective and selective inhibitory reagents, elucidate their precise in vivo mechanism of action, and combine these molecules with other anti-cancer strategies in a logical way.

## Figures and Tables

**Figure 1 cancers-13-04529-f001:**
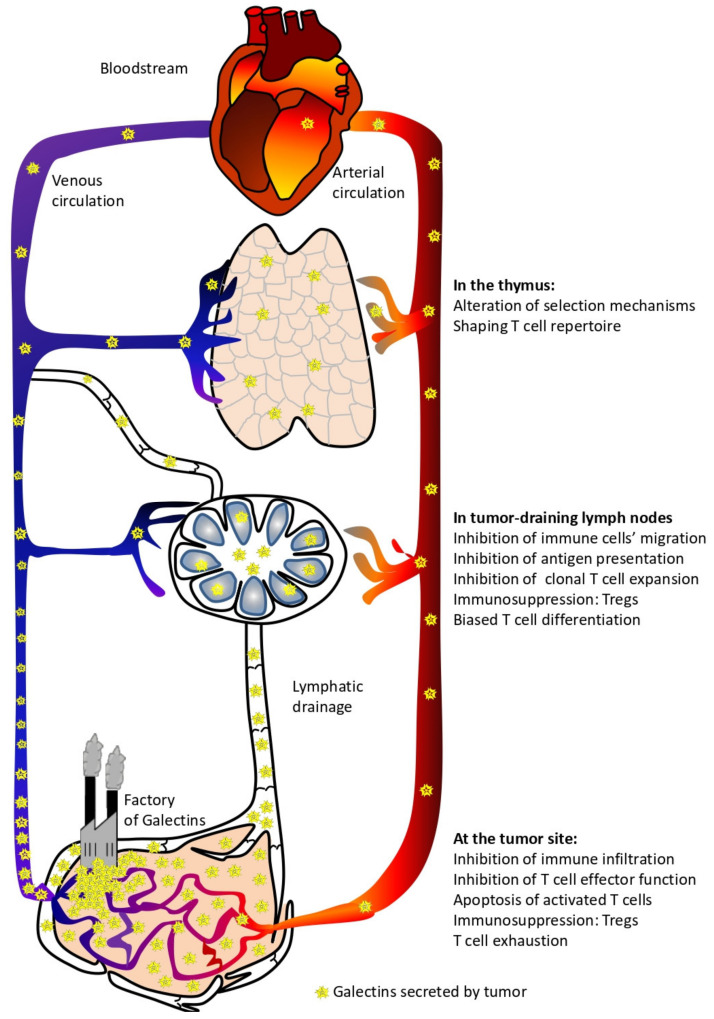
Local and systemic effects of circulating tumor-derived galectins.

**Table 1 cancers-13-04529-t001:** The galectins’ interactors and their impact on cancer immune escape.

Member	Recognition Motif	Interactors	Described Biological Effects	References
**Galectin-1**	**Carbohydrate-dependent** Long poly-N-acetyllactosamine chains with a terminal β-galactose residue **Carbohydrate-independent**	**Proteins of the extracellular matrix:** Laminin, fibronectin, collagen, vitronectin, thrombospondin, osteopontin **Membrane receptors:** Neuropilin-1, VEGF-R2, integrins α1β1 and αMβ2, actin, CD43, CD3,CD4, CD2, CD7, and CD45 Protocadherin-24 GM1 H-RAS Gemin-4, Transcription Factor II-I (TFII-I), snRNP	Involved in ligand crosslinking and the formation of cell surface lattices Induction and maintenance of multiple signaling pathways Modulation of cell adhesion and migration Modulation of vascular function and immune cell migration Modulation of T cell activation, survival and acquisition of effector functions Regulation Wnt signaling Modulation of Treg function Enhance signaling through GTP-H-RAS nanoscale signaling hubs Nuclear splicing of pre-mRNA and control of gene expression	[43,44] [45,46,47] [48,49,50] [51,52,53,54] [55,56,57,58,59,60] [61] [62,63] [64,65] [66,67,68,69,70]
**Galectin-3**	**Carbohydrate-dependent** Repeating [-3Galβ1-4GlcNAcβ1-]n or poly-N-acetyllactosamine sequences regardless of the presence of a terminal β-galactose residue **Carbohydrate-independent**	**Proteins of the extracellular matrix:** Laminin, vitronectin, collagens I and IV **Soluble cytokines** **Membrane receptors:** TCR complex, CD45, CD71, LFA-1, MCAM, TLR-4 LAG-3, VEGF-R2 Nuclear mitotic aparatus (NuMa) K-RAS β-catenin, Protocadherin-24 Endosomal sorting complex required for transport (ESCRT) or Alix Centrin-2 in basal bodies and centrosomas Synexin, CD95 (APO-1/FAS) Cyclin D1/CDK4 complex, hTERT, ATP synthase Gemin-4, snRNP, transcription factor II-I (TFII-I)	Involved in ligand crosslinking and the formation of cell surface lattices Impact on induction and maintenance of multiple signaling pathways: cell adhesion and migration Reduces cell migration by trapping cytokines Modulation of T cell migration, activation, and functional secretory synapse Cell division Determine membrane nanostructure: impact on signal transduction Wnt signaling TCR downregulation, EGFR trafficking, biogenesis of multivesicular bodies-Exosomes Microtubule organization Apoptosis Cell cycle and senescence Nuclear splicing of pre-mRNA and control of gene expression	[43,44] [50,71,72] [72] [73,74,75,76,77,78,79,80] [81] [82] [61,83,84] [85,86,87,88,89] [90] [91,92] [93,94,95] [67,68,69,70,96,97]
**Galectin-7**	Internal or terminal LacNAc repeats	Tid1 Bcl-2 Smad 3	Tid1 regulates the nuclear translocation of Galectin-7: role in tumorigenesis and metastasis Sensitize mitochondria to apoptosis signals Decrease expression of TGF-β responsive genes	[98] [99] [100]
**Galectin-8**	Human blood groups A and B glycans and sialylated lactose or lactosamine	**Proteins of the extracellular matrix:** Laminin, fibronectin, vitronectin, collagen IV Integrins (α3β1 and α6β1), CD166, podoplanin, CD44 NDP52	Modulation of cell adhesive and signaling properties Regulation of cell adhesion to endothelium and migration Regulation of apoptosis and inflammation Regulation of bacteria-specific autophagy	[50,101] [102,103,104,105] [106] [107,108]
**Galectin-9**	Poly N-acetyllactosamine units	TIM-3, PD-1, CD44, VISTA, 4-1BB, CD40, DR3 Cell surface protein disulfide isomerase, β3 integrin IgE Glut-2 NF-IL6 transcription factor	Regulation of T-helper 1 cell immunity and tolerance induction Regulation of the redox environment and T cell migration Anti-allergic effects Determines Glut-2 cell-surface half-life, metabolism regulation Regulation of inflammatory cytokines	[109,110,111,112,113,114,115,116,117] [118] [119] [120] [121]

**Table 2 cancers-13-04529-t002:** The main inhibitors of galectins evaluated for cancer treatment.

Member	Inhibitor	Reported Biological Effects	Phase of The Study	References
**Galectin-1**	Thiodigalactoside (TDG)	Inhibition of tumor growth and metastasisReduction of angiogenesisActivation of anti-tumor immunosurveillance	Pre-clinic	[380,381]
	Anginex (βpep-25)	Inhibition of tumor growth, angiogenesis and migrationIncrease sensitivity to radiotherapy, chemotherapy and anti-angiogenesis therapy	Pre-clinic	[20,254,313,341,382,383]
	Anginex analogues (6DBF7; DB16; DB21)	Inhibition of tumor growth and angiogenesis	Pre-clinic	[384,385]
	OTX008	Inhibition of tumor growth, angiogenesis and migrationSynergic effects with chemo- and immunotherapy	Pre-clinicPhase I	[315,359,386,387]NCT01724320
	Galectin-1–specific neutralizing mAb	Inhibition of tumor growth and angiogenesis	Pre-clinic	[25,52,388,389]
		Increased immune infiltration of tumors		
**Galectins-1 and -3**	GM-CT-01 (DAVANAT^®^)	Inhibition of tumor growthRestoration of anti-tumor immune surveillance	Pre-clinicPhase I and II	[390]NCT00054977,NCT00388700,NCT00386516,NCT01723813
	GR-MD-02(modified version of the DAVANAT®)	Inhibition of tumor growth, improve survival of tumor-bearing miceRestoration of anti-tumor immune surveillance, improve immune checkpoint blockade	Pre-clinicPhase I	[365,391]NCT02117362,NCT02575404
**Galectin-3**	G3–C12	Inhibition of tumor growth and mestastasisSynergic effects with chemotherapy	Pre-clinic	[392,393,394,395,396]
	Modified citrus peptin (MCP)	Inhibition of tumor growth, angiogenesis and metastasis	Pre-clinic	[397,398,399,400,401,402]
		Immune activation		
		Increased sensitivity to chemotherapy		
	PectaSol-C Modified Citrus Peptin	Synergic effects with chemotherapy	Pre-clinic	[331,403,404]
			Phase III	NCT01681823
	GCS-100	Inhibition of tumor growth	Pre-clinic	[265,327,405,406,407,408]
		Correction of impaired anti-tumor immune functions	Phase I	NCT00609817
		Increased sensitivity to Immuno-chemotherapy	Phase II	NCT00514696,NCT00776802
**Galectin-7**	Inhibitory peptide	Inhibition of tumor growth and metastasis	Pre-clinic	[370]
		Restoration of anti-tumor immune surveillance, improve immune checkpoint blockade

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
