# Peer review of "Unraveling How Tumor-Derived Galectins Contribute to Anti-Cancer Immunity Failure"

_cancers, 2021, doi:10.3390/cancers13184529_

Round 1
Reviewer 1 Report
This review is on a highly important topic and should be of potential interest. Unfortunately, the narrative is difficult to follow due to numerous grammatical mistakes and poor style. In my view, the manuscript requires substantial revision and careful editing in terms of English grammar and scientific style (see examples below). The manuscript would also benefit from restructuring as well as summarizing the main properties and functions of the various galectins in a figure or table, and their differential roles in various cancers. Regarding galectin-9, while the authors briefly mention its association with Tim-3 and PD-1, it is now known to form complexes with VISTA too. I think they should more comprehensively address the biological functions of galectins, their binding partners, and include more details regarding their mechanism of action in terms of causing immune escape.
Minor points
The “simple summary” need rewriting since there are several grammatical mistakes and the style is difficult to follow.
Do not capitalize the first letter for galectins within a sentence.
Line 129: poor style “While a little bit more potent….”
Line 153 (and elsewhere): poor style “One might wonder if …”
Line 554: INFγ-R2…should be rewritten as IFNγ-R2
Line 613: “Galectin-9 is a potent mean to deactivate these cells..” Please rewrite
Line 626: “Tregs are very sensible to Galectin-9 levels…” Sensitive not sensible
Author Response
This review is on a highly important topic and should be of potential interest.
We thank the reviewer for pointing out the relevance of the review’s topic.
Unfortunately, the narrative is difficult to follow due to numerous grammatical mistakes and poor style. In my view, the manuscript requires substantial revision and careful editing in terms of English grammar and scientific style (see examples below).
We apologize for the mistakes indicated by the reviewer in minor points. Such mistakes were corrected in the revised version. In addition, the manuscript was proofread by an English-native scientist. We believe these changes have clarified the message of the manuscript; we thank the reviewer for such a comment.
The manuscript would also benefit from restructuring as well as summarizing the main properties and functions of the various galectins in a figure or table, and their differential roles in various cancers. Regarding galectin-9, while the authors briefly mention its association with Tim-3 and PD-1, it is now known to form complexes with VISTA too. I think they should more comprehensively address the biological functions of galectins, their binding partners, and include more details regarding their mechanism of action in terms of causing immune escape.
Thanks to the reviewer for this suggestion that improves the clarity of the message. The revised manuscript included a new Table I summarizing galectins' main binding partners and functional properties (focusing on those with immune relevance in cancer). In particular, we have included details about the interaction VISTA-galectin-9, as suggested by the reviewer (page11, line 439; page 16, line 673 and Table I).
Minor points
The “simple summary” need rewriting since there are several grammatical mistakes and the style is difficult to follow.
The simple summary was completely rewritten. Thanks to the reviewer for this suggestion that improves the summary.
Do not capitalize the first letter for galectins within a sentence.
Throughout the text, Galectins was changed to galectins.
Line 129: poor style “While a little bit more potent….”
This phrase was changed (page 5 line 136): Further, galectin-3 is more potent at inducing cell apoptosis when compared to galectin-1. This reviewer suggestion improves the manuscript with more scientific terminology.
Line 153 (and elsewhere): poor style “One might wonder if …”
These sentences were changed as follows:
Page 5 line 101: In this context, do galectins produced by the multiple cell types within a tumor impact T cell production and 102 education in the thymus?
Page 6 line 162: Thus, galectins produced abundantly by tumors could shape the repertoire of newly generated T lymphocytes.
Line 554: INFγ-R2…should be rewritten as IFNγ-R2
The typographic mistake was corrected (page 14, line 573).
Line 613: “Galectin-9 is a potent mean to deactivate these cells.” Please rewrite
This phrase was changed (page 15 line 648) as follows: Intratumor CD103+ dendritic cells express Tim-3; its interaction with galectin-9 induces the deactivation of antigen-presenting cells.
Line 626: “Tregs are very sensible to Galectin-9 levels…” Sensitive not sensible
The mistake was corrected (page 15, line 662 and page 16, line 676).

Reviewer 2 Report
Summary: This is an interesting review on the role of galectins in cancer. The authors address several issues that are rarely discussed in publications or reviews on galectins. This include a discussion on the use of recombinant galectin in in vitro experimental systems, the role of oxidation for galectin-1, etc. It also includes a well-documented discussion on the potential role(s) of galectins in the thymus during the differentiation and selection of T cells. I congratulate the authors for addressing these issues.
Major comments :
This review aims to deepen our knowledge on how Galectins, abundantly produced by tumors, can impact on the production, activation and effector function of anti-tumor T lymphocytes. Yet, the rationale for focusing of gal-1, -3, -8 and -9 is not clear as other galectins are also well known to (down) regulate the immune response. I certainly not agree with the comment that these galectins are the main galectins produced by tumors. This is a simplistic view as the repertoire of galectins depend on the tumor type and/or subtypes. In my opinion, the major omission is gal-7. There are more than 80 Pubmed articles on the role of gal-7 in cancer, with more than a third on the regulation of the immune response by gal-7. This is more than papers published on galectin-8, which role has not been investigated deeply as acknowledged by the authors on page 9 of the review. Moreover, gal-7 is expressed at abnormally high levels in many cancer types and often correlates with poor survival and cancer aggressiveness. This is the case for cancer of the esophagus, breast cancer, etc. This has been reported by many groups, including a recent paper published in Cancers [Trebo et al., 2020]. There have also been considerable interest to develop gal-7 inhibitors. Some of these inhibitors have also been shown to inhibit gal-7-induced apoptosis of human T cells. I believe that the authors should include gal-7 in their review and comment, even briefly, on the role of other galectins.
The quality of figure 1 is average. Considering the detailed description of the role of galectins in thymic development, I would have included a higher quality picture that better summarizes the role of each galectin within the thymus and their cellular localization, focusing for example on specific stromal compartments (cortical versus medulla, for example). This would considerably add to the originality of the review.
The authors should be prudent in their conclusion that tumors « produce » galectins. It is important to remind the readers that galectins are not only secreted by cancer cells, but also by stromal cells of the tumor environment and by infiltrating immune cells. Galectins produced by these cells (and not only by tumors cells as written by the authors) certainly contribute to the modulation of immune cell filtration. This issue does not come out sufficiently clear in my opinion.
I’m surprised that the authors did not cite the paper by Perillo et al published in Nature in 1995. This paper was the first one showing that role of galectin-1 in killing activated T cells. They were also the first one to propose that gal-1 may regulate the immune response in cancer. This was 6 years before the publication of Van der Brûle in 2001, as cited by the authors.
The section and discussion on galectin inhibitors could be improved as it lacks criticisms on past and current efforts on the development of galectin inhibitors. Many of the cited studies in Table I for example, have been published 10-15 years ago. Yet, none of them has met with clinical success. The reasons for this are numerous. It would have been helpful for the readers to have the opinion of the authors on why it is the case from an immunological point of view. What are the challenges associated with the development on galectin inhibitors? Why did they fail? What is the future? What are the most promising avenues?
Minor comment :
Although the English used is correct and readable, overall, the quality of the writing could be improved. The flow of the text is arduous.
Author Response
Summary: This is an interesting review on the role of galectins in cancer. The authors address several issues that are rarely discussed in publications or reviews on galectins. This include a discussion on the use of recombinant galectin in in vitro experimental systems, the role of oxidation for galectin-1, etc. It also includes a well-documented discussion on the potential role(s) of galectins in the thymus during the differentiation and selection of T cells. I congratulate the authors for addressing these issues.
We thank the reviewer for pointing out the relevance of the review’s topic.
Major comments :
This review aims to deepen our knowledge on how Galectins, abundantly produced by tumors, can impact on the production, activation and effector function of anti-tumor T lymphocytes. Yet, the rationale for focusing of gal-1, -3, -8 and -9 is not clear as other galectins are also well known to (down) regulate the immune response. I certainly not agree with the comment that these galectins are the main galectins produced by tumors. This is a simplistic view as the repertoire of galectins depend on the tumor type and/or subtypes. In my opinion, the major omission is gal-7. There are more than 80 Pubmed articles on the role of gal-7 in cancer, with more than a third on the regulation of the immune response by gal-7. This is more than papers published on galectin-8, which role has not been investigated deeply as acknowledged by the authors on page 9 of the review. Moreover, gal-7 is expressed at abnormally high levels in many cancer types and often correlates with poor survival and cancer aggressiveness. This is the case for cancer of the esophagus, breast cancer, etc. This has been reported by many groups, including a recent paper published in Cancers [Trebo et al., 2020]. There have also been considerable interest to develop gal-7 inhibitors. Some of these inhibitors have also been shown to inhibit gal-7-induced apoptosis of human T cells. I believe that the authors should include gal-7 in their review and comment, even briefly, on the role of other galectins.
We thank the reviewer for pointing out the relevance of other galectins in cancer, in particular galectin-7. As the reviewer stated, galectin-7 is up-regulated in different types of cancers. Accordingly, we have added reported data about the role of this galectin in central and peripheral immune behavior, which could impact cancer immune escape (page 7, line 210; page 14, line 606; and Table II includes a galectin-7 inhibitor).
The quality of figure 1 is average. Considering the detailed description of the role of galectins in thymic development, I would have included a higher quality picture that better summarizes the role of each galectin within the thymus and their cellular localization, focusing for example on specific stromal compartments (cortical versus medulla, for example). This would considerably add to the originality of the review.
This point is probably the only one we respectfully discern with the reviewer. In fact, we cite the anatomical compartmentalization of some thymic functions that had been described to be regulated by galectins (apoptosis and cell-to-cell interactions). However, we developed arguments refuting that circulating tumor-derived galectins might be implied in such functional effects. On the contrary, we believe circulating galectins have other biological functions in the thymus, which is one of the main conclusions of our review. Currently, there is no report available describing the anatomical compartmentalization for these later functions. Therefore, we decided to keep the figure as initially presented.
The authors should be prudent in their conclusion that tumors « produce » galectins. It is important to remind the readers that galectins are not only secreted by cancer cells, but also by stromal cells of the tumor environment and by infiltrating immune cells. Galectins produced by these cells (and not only by tumors cells as written by the authors) certainly contribute to the modulation of immune cell filtration. This issue does not come out sufficiently clear in my opinion.
We agree with the reviewer. Accordingly, we have emphasized galectins’ source in tumors (transformed cells and stroma) (page 2, line 56; page 5, line 101; page 12, line 462; and page 14, line 612).
I’m surprised that the authors did not cite the paper by Perillo et al published in Nature in 1995. This paper was the first one showing that role of galectin-1 in killing activated T cells. They were also the first one to propose that gal-1 may regulate the immune response in cancer. This was 6 years before the publication of Van der Brûle in 2001, as cited by the authors.
We believe there was a misunderstanding in this respect. Perillo et al (1995) was cited in the initial version of our manuscript (page 11, line 466; page 11, line 469 and page 4, line 112).
Van der Brûle was included at the beginning of the tumor-related chapter because this author was the first to propose the “tumor protective shield” concept. However, we entirely agree that this concept was proposed 6 years later the first experimental demonstration of galectin-1 as a T cell pro-apoptotic factor. In the revised version of the manuscript, Perillo et al. reference appears next following the concept, and to further stress its novelty at that time, the phrase was reformulated as follows (page 12, line 483):
Indeed, a seminal article published in 1995 was the first to demonstrate that galectin-1 induces apoptosis of T lymphocytes [139]. While this pioneering result showed that all T lymphocytes bind galectin-1 [139];…
The section and discussion on galectin inhibitors could be improved as it lacks criticisms on past and current efforts on the development of galectin inhibitors. Many of the cited studies in Table I for example, have been published 10-15 years ago. Yet, none of them has met with clinical success. The reasons for this are numerous. It would have been helpful for the readers to have the opinion of the authors on why it is the case from an immunological point of view. What are the challenges associated with the development on galectin inhibitors? Why did they fail? What is the future? What are the most promising avenues?
We greatly appreciate the reviewer for this constructive comment. The manuscript was modified accordingly. A new chapter was included referring to the problem of galectin inhibitors and the challenges their use implies (page 20, line 877).
Minor comment :
Although the English used is correct and readable, overall, the quality of the writing could be improved. The flow of the text is arduous.
The manuscript was proofread by an English-native scientist. We believe this correction has clarified the message.

Round 2
Reviewer 2 Report
All comments were adequately addressed.